# Corn Starch: Quality and Quantity Improvement for Industrial Uses

**DOI:** 10.3390/plants11010092

**Published:** 2021-12-28

**Authors:** Ju-Kyung Yu, Yong-Sun Moon

**Affiliations:** 1Syngenta Crop Protection LLC, Seeds Research, Research Triangle Park, Durham, NC 27709, USA; ju-kyung.yu@syngenta.com; 2Department of Horticulture & Life Science, Yeungnam University, Gyeongsan 38541, Korea

**Keywords:** corn, starch, starch biosynthesis, wet milling, amylose, amylopectin, waxy, amylose-extender

## Abstract

Corn starch serves as food, feed, and a raw material for industrial use. Starch makes up most of the biomass of the corn hybrid and is the most important and main yield component in corn breeding programs. Starch is composed of two polymers, branched amylopectin and linear amylose, which normally constitute about 75% and 25% of the corn starch, respectively. Breeding for corn starch quality has become economically beneficial because of the development of niche markets for specialty grains. In addition, due to the increased demands of biofuel production, corn ethanol production is receiving more attention. Consequently, improving starch quantity has become one of the most important breeding objectives. This review will summarize the use of corn starch, and the genetics and breeding of grain quality and quantity for industrial applications.

## 1. Introduction

Corn is the third most important grain crop in the world after wheat and rice. It serves as a primary energy source in livestock feed. Corn is processed into a multitude of food and industrial products, including starch, sweeteners, corn oil, beverages and industrial alcohol, and biofuel (Figure 1). In the US, roughly 40% of corn croplands are used for ethanol production and another 40% are used for livestock feed (Figure 2). 

Starch is a major component of corn and makes up approximately 75% of its dry matter, which accumulates in the endosperm tissue [1,2,3]. Starch is utilized in various industrial applications, and about 1.6% of US corn is used for these purposes (USDA National Agricultural Statistics Service: https://www.nass.usda.gov/ accessed on 27 November 2021). Due to the premiums offered by end-users, specialty starch corns are receiving more attention as potentially profitable alternatives to conventional field corn. Traditionally, corn has been treated and marketed as a bulk commodity. Corn grain users and processors have become more interested in the quality characteristics of corn grain. Corn that meets specific user needs is called value-added corn. For example, waxy corn for wet milling is usually grown under contract for wet-corn millers with a premium of USD 0.55 per bushel (US Grains Council: https://grains.org/ accessed on 27 November 2021). In another example, the Enogen corn variety provides alpha amylase enzymes to ethanol plants, and ethanol producers are willing to reward farmers with a premium of up to USD 0.40 per bushel (US Grains Council: https://grains.org/ accessed on 27 November 2021).

Plant breeders have modified the starch of corn to better meet the needs of the livestock feeder, food industry, and industrial users [1,2,3,4]. The purpose of this review is to summarize the use of corn starch, and the genetics and breeding of starch quantity and quality for industrial applications.

## 2. Wet Milling Process

Starch is typically produced by the wet milling process [5,6]. Wet milling, which excludes ethanol production, uses 10% of corn harvested in the US to make starch and sweeteners and is one of the prime product outputs in addition to oil, protein, gluten meal, and fiber (National Corn Growers Association: https://www.ncga.com/ accessed on 27 November 2021). In wet milling, the grain is soaked or steeped in water and dilute sulfurous acid for 24 to 48 h. This steeping facilitates separation of the grain into many of its component parts. After steeping, the corn slurry is processed to separate the corn germ. The corn oil from the germ is extracted, and the remaining starch, gluten, and fiber components are further segregated. The starch can then be processed in one of three ways: fermented into ethanol, dried, and sold as dried or modified corn starch, or processed into corn syrup (Figure 3).

The conventional wet-milling process is a capital-, time-, and energy-intensive process [5,6]. For optimal starch recovery, the steeping of corn kernels is typically conducted for 24–48 h and requires roughly 21% of the overall plant energy and capital costs. It is a water-intensive technology as 1.5 m^3^ of fresh water per ton of corn is needed. It additionally involves chemical steeping in a solution of sulfurous acid (SO_2_ in water), which, according to the US Environmental Protection Agency (https://www.epa.gov/ accessed on 27 November 2021), is one of the six most common air pollutants and results in negative effects on the environment. A few advanced wet-milling technologies have been introduced, such as enzymatic wet milling to shorten steeping time and reduce SO_2_ application, saving 5% of operation costs, and ultrasound-assisted wet milling, which increases starch yield by 10% with the same input resources [7,8]. In addition to the improvement in the wet-milling process, starch can be modified genetically to reduce the cost of production or increase the financial benefit of end-users. Breeding objectives are set to improve overall starch quantity or produce starch with novel physical properties (quality), and achieving these would benefit farmers and end-users. 

## 3. Starch Biosynthesis

Corn starch is composed of two polymers (homopolysaccharides), amylopectin and amylose, which differ in their chain length and degree of branching. Amylopectin is more highly branched (a chain of α-D-(1-4) and α-D-(1-6)-glucosidic linkages) and normally constitutes about 75% of the starch granule. Amylose is mostly linear (α-D-(1-4)-linked glucose residues) and constitutes 25% of the starch granule [9,10,11,12,13]. These two polymers have different physical properties, and the ratio of amylose to amylopectin plays a key role in the appearance, structure, and quality of industrial processing. Therefore, much of the effort of breeding for distinctive starch properties has involved changing the relative proportions of these two polymers.

A wide range of research has been conducted to understand the genetics and biochemistry of starch synthesis. The simplified starch biosynthesis pathway with key enzymes is illustrated in Figure 4. Starch biosynthesis in corn endosperm begins with the cleavage of sucrose into fructose and UDP-glucose, catalyzed by sucrose synthase (SUS) encoded by sucrose synthase (*sh1*). Following this, the products are converted into ADP-glucose (ADPG) by ADP-glucose pyrophosphorylase (AGPase). AGPase is encoded by brittle2 (*bt2*, the small subunit) and shrunken2 (*sh2*, the large subunit). Amylose and amylopectin use ADPG as the activated glucosyl donor for synthesis, but they are produced by different enzymes subsequently. Granule-bound starch synthase (GBSS, encoded by waxy (*wx*)) is solely responsible for amylose synthesis. Amylopectin biosynthesis requires three coordinated enzymes, including starch-branching enzyme (SBE, encoded by amylose extender1 (*ae1*)), debranching enzyme (DBE, encoded by sugary1 (*su1*)) and starch synthase (SS), to elongate the amylopectin chain [3,9,10,11,12,13].

A lack of the SBE enzyme in the *ae1* gene drives less amylopectin production, resulting in the accumulation of up to 50% amylose. In addition, amylopectin depletion has been achieved with *ae1* and *su1* mutants [14]. Mutation of the *waxy* gene abolishes amylose synthesis, resulting in 100% amylopectin in the endosperm [15]. Therefore, these mutants have been utilized in corn breeding to modify the ratio of amylose to amylopectin and improve starch quality (Table 1). 

## 4. Starch Quality Improvement: High Amylopectin (Waxy Corn and Glutinous Corn)

Waxy corn is a starch variant carrying a mutant allele of *Waxy1* (*Wx1*), a single recessive gene (Table 1). The *Waxy1* gene is required for the biosynthesis of amylose, and the mutant alleles of *Wx1* produce waxy starch that is comprised of 100% amylopectin [15,16]. Waxy corn is mostly produced under contract for starch companies. A premium is paid for this as compensation for the extra costs incurred from the lower yield (about 5% less than its non-waxy corn counterpart) and the extra handling involved, such as quality control procedures to ensure starch in the grain is not contaminated. Waxy corn is processed in wet milling to produce waxy starch, which is relatively easy to gelatinize, and it produces a clear viscous paste with a sticky or tacky surface. Therefore, it is mainly used by food companies as a stabilizer and thickener. Moreover, it is used in the textile, adhesive, corrugating, and paper industries.

Corn with a *wx1* mutant allele was discovered in the early 1900s [15]. A single recessive gene (*wx*) is located on the short arm of chromosome 9 and encodes for the waxy endosperm of the kernel, while the *Wx* gene encodes for endosperm in normal starch. The structure of the wildtype waxy locus has been sequenced (Gene ID: 541854), and the gene has 3676 bp composed of 14 exons. Globally, there are more than 50 natural mutant alleles in the *wx* locus identified at the molecular level, and diverse research continues to characterize new alleles in waxy locus because of its high economic value [16]. Recently, new mutant alleles, *wx-hAT*, were identified, consisting of a 2286 bp transposon inserted into the middle of exon three of the waxy gene [17]. 

The breeding of waxy corn is relatively straightforward as waxy starch phenotyping is simple: the kernels stain reddish-brown when treated with potassium iodide, whereas normal starch stains blue. The backcross breeding has been used extensively to introgress waxy mutant alleles into the elite lines and convert the best hybrids to the waxy phenotype. However, mutant alleles are recessive, which can make the breeding operation cumbersome, requiring substantial time and resources. In addition, there is often a yield penalty associated with introgression of the waxy alleles by the linkage drag effect [16,17]. 

To ensure breeding efficiency, the CRISPR-Cas9 system is being adopted. Qi et al. [18] created target mutations of the *Wx* locus in the ZC01 background (ZC01-DTMwx) using the CRISPR-Cas9 system. A total of six mutants were obtained among progeny crossed with ZC01-DTMwx. The mutant lines reached 94.9% of amylopectin contents in the endosperm starch compared to wild-type controls, while the agronomic performance remained similar. The researchers applied triple selection to segregants to achieve high background genome recovery with transgene-free *wx* mutations and to overcome the yield penalty. In addition, Gao et al. [19] generated waxy corn hybrids by CRISPR–Cas9 editing of two waxy deletion alleles, 4 kb and 6 kb deletions in 12 elite inbred lines. CRISPR-wx hybrids reached higher than 95% of amylopectin content. Field trials at 25 locations demonstrated approximately 5.5 bushels per acre higher yields than conventional backcrossing-introgressed waxy hybrids. This demonstrated that the direct genome-edited hybrids reduce the impact of linkage drag on yield.

## 5. Starch Quality Improvement: High Amylose (Amylomaize)

Normal corn generally contains an amylose content of roughly 25% that of total starch. Amylomaize is the generic name for corn that has an amylose content greater than 50% [9,10]. The single recessive amylose-extender (*ae*) gene with modifiers increases the ratio of amylose to amylopectin in starch (a range of 50 to 94% amylose content), but it does not change the total starch amount in the grain (Table 1) [14]. Similar to waxy corn, high-amylose corn is grown under contract for wet milling with premiums of higher than USD 1 per bushel. The two major types of high-amylose corn are commercialized: Class V (amylose percentages in the 50% range) and Class VII (amylose content from 70 to 80%) (https://www.hort.purdue.edu/newcrop/articles/corntypes.html accessed on 27 November 2021). The starch from high-amylose corn is used in textiles, candies, and adhesives. Recently, the interest in high amylose starches has increased for two purposes: (1) starch-based biodegradable thermoplastics, such as shopping bags, and packing material, and (2) as a source of resistant starch (RS), a type of starch that resists digestion. As a food additive, consumers can benefit from added RS, since recent research in food science has validated that it lowers the glycemic index and the risk of colon cancer [20,21].

The single recessive gene mutant amylose-extender (*ae*) is located on chromosome 5, which encodes the starch-branching enzyme II (*SBEIIb*) (Table 1) [14,22,23,24,25]. This decreases the total activities of SBEs and increases the amylose proportion up to 60%. The wild-type *ae* locus has been sequenced (Gene ID: 542238), and the gene has 16,939 bp composed of 22 exons. Unlike waxy corn, *ae* mutant alleles do not confer 100% amylose starch, but rather a range of amylose proportions depending on genetic modifiers (multi-enzyme reaction) and the genetic background and environment (cooler temperature results in higher amylose content) [14,22,23]. The well-known gene mutant amylose extender (*ae*) has been identified to produce high-amylose corn lines (AE11 line) since 1948 [26]. However, there is still a significant shortage of high-amylose corn due to the limitation of natural germplasm resources. Campbell et al. [27] released GEMS-0067 (PI 643420) publicly, which is an elite, amylomaize Class VII line derived from the crossing of GUAT209:S13 and (H99ae × OH43ae). It possesses high amylose modifier genes that, together with the recessive amylose-extender (*ae*) allele, result in an increase in starch amylose content higher than 70%.

Wu et al. [28] studied the gene effects, non-allelic interactions, and heritability of high amylose content in GEMS-0067 lines using nine populations that were derived from a cross between H99ae, a corn inbred line with 55% amylose starch, and GEMS-0067. The segregants’ inheritance was analyzed by a triploid endosperm model. An additive effect was found as a main contributor to amylose content. Incomplete dominance and maternal effects also explained some of the inheritance of high amylose starch, and maternal effects were also detected. The encouraging findings for breeders in both broad-sense and narrow-sense heritability were of high quality; thus, high amylose content could be effectively selected for in a segregating population. Further genetic research was conducted by Han et al. [29] to characterize the GEMS-0067 line at the molecular level. The deletion of the ninth exon of the *SBEIIb* gene containing 84 bp decreased the total activities of SBEs to about 71% and increased the amylose proportion of corn kernels up to 60%. The high amylose proportion suggested that this germplasm line was derived from the same resource of the high amylose inbred line, AE11. In general, there is a negative correlation between amylose proportion and starch content that was found in most of the high amylose inbred germplasm lines. However, this study showed that the kernel weight per ear of GEMS0067 was only 8.3% lower than its wild type. Therefore, GEMS0067 was recommended as a high potential elite germplasm line for corn breeding of a high amylose proportion. 

SBEs have multi-isoforms in corn; *SBEIIa* and *SBEIIb* play a key role in amylopectin synthesis. In kernels of the *SBEIIa* mutant, the degree of starch branching and composition are unaffected [16,23,25,26]. However, in kernels of the *SBEIIb* mutant (the *ae* mutant), kernels contain elongated starch granules, exhibit drastically increased amylose content, and the structure of amylopectin is distinctly altered, exhibiting a lower branching degree [21,23,24,25,30]. Zhao et al. [31] reported the successful application of RNAi technology for improving amylose content in corn endosperm through the suppression of the *ZmSBEIIa* and *ZmSBEIIb* genes by hairpin SBEIIRNAi constructs into the elite inbred corn line Chang7–2. These SBEIIRNAi transgenes led to the down-regulation of *ZmSBEIIa* and *ZmSBEIIb* expression and SBE activity to various degrees and altered the morphology of starch granules. Transgenic corn lines with amylose content of up to 55.9% were produced, and these avoided the significant decreases in starch content and grain yield that occur in high-amylose *ae* mutants.

## 6. Starch Quantity Improvement

Currently, the fastest growing use of corn is ethanol fuel production. Ethanol fuel production is making a significant contribution to creating renewable fuels and reducing carbon emissions for a cleaner environment [32,33]. One important breeding objective to support the ethanol production industry is to increase the total amount of extractable starch during wet and dry milling [32].

There are various researchers attempting to enhance total starch accumulation and a consequent increase in starch quantity. One example involves engineering ADP-glucose phosphorylase (AGPase) activity in endosperm. AGPase is composed of two subunits, encoded by the Brittle2 (*Bt2*) gene and the Shrunken2 (*Sh2*) gene (Table 1). Li et al. [34] generated transgenic plants that are an over-expression of both the *Sh2* and *Bt2* genes using an endosperm specific promoter. The over-expression of both genes enhanced AGPase activity, resulting in starch content increasing to over 74% compared with the wildtype starch content of 65%. Another intriguing example explores the function of Dof (DNA binding with one finger) transcription factor expressed during kernel development. Qi et al. [35] discovered endosperm-specific Dof protein (ZmDof3) in corn using whole genome-wide RNA expression profiling. The research explained that the expression level of *ZmDof3* is associated with a starch-synthesis-related gene, such as *Su2* (encodes SSIIa). *ZmDof3* knockdown prohibited starch accumulation, resulting in reduced starch content.

As previously addressed, starch biosynthesis is a complex trait controlled by many genes and modifiers; thus, there are multiple potential breeding targets for total starch improvement. Jiang et al. [36] demonstrated a multigene engineering approach: over-expression of *Bt2*, *Sh2*, *Sh1* and *GBSSIIa* (to enhance the activity of sucrose synthase (SS), AGPase, and granule-bound starch synthase (GBSS)), with the suppression of SBEI and SBEIIb (to reduce the activity of starch branching enzyme) using RNAi technology. Transgenic corn plants expressed all six genes, which increased endosperm starch content from 2.8% up to 7.7%, and there was a 37.8–43.7% increase in the proportion of amylose compared to untransformed control plants. Additional agronomic trait improvements were found, such as a 20.1–34.7% increase in 100-grain weight, and a 13.9–19% increase in ear weight. These promising results that unlock the multigene engineering approach can be applied to improve starch quantity and quality as well as starch-dependent agronomic traits simultaneously.

In addition to the genetic engineering approaches addressed above, conventional breeding approaches with DNA markers have also been performed. With advanced genotyping technology and computational biology, the genome-wide association study (GWAS) is becoming a popular method to characterize the causal relationship between genetic polymorphisms (gene) and biological traits (phenotype). By conducting a GWAS in a set of 263 inbred lines, Liu et al. [37] identified four significant SNPs and 77 candidate genes, including a key gene of glucose-1-phosphate adenylyltransferase (AGP), which regulates starch biosynthesis in corn kernels. These SNPs in candidate genes could be used to select high starch content lines in the DNA marker-assisted breeding program to improve germplasm. 

## 7. Perspectives

We understand the key steps in starch metabolism, but how they are connected requires further clarification. The molecular mechanism of network and regulation involved in overall starch biosynthesis and accumulation remains unclear. From a breeding perspective, single-gene mutants are used to give stepwise improvements in these traits. As addressed in this review, there are many genes and modifiers that contribute to starch biosynthesis. We continue to face challenges in starch breeding of corn [38].

Throughout the past decades, there have been revolutionary advancements in various omics and sequencing technologies, as well as computational biology. Proteomics datasets have been produced to study the characterization of the coordinated accumulations of numerous proteins responsible for starch biosynthesis in corn (reviewed in Niu et al. [39]). DNA marker-based backcrossing breeding programs have improved starch traits by facilitating the transfer of mutant alleles [40]. However, the potential of this type of breeding program for improving complex traits is limited. The third-generation sequencing technologies not only contribute to increasing genome sequence information but also enrich genotype data (GBS) to facilitate genomic selection (GS). GS exploits genome-wide information and has become increasingly relevant to continuous germplasm improvement and improving the rate of genetic gain, which can overcome the limitation of backcross breeding [41]. Today, it is a routine breeding process in various disease resistance breeding and yield breeding programs regarding corn. Authors believe it might provide useful breeding technology to improve starch quality and quantity for complex traits such as starch, which is controlled by several genes with modifiers and significant environment interaction [42,43].

Moreover, other challenges have begun to surface with large-scale and multi-dimensional datasets produced by advanced high-throughput genomic, transcriptomic, proteomic data, and phenotype and environmental data. Regarding these, machine learning (ML) plays a pivotal role in data mining and analysis, providing relevant information that can be used for efficient interpretation of results and decision making toward characterizing the gene network and achieving breeding targets [44,45]. ML has rapidly evolved and is now widely applied in genetic research and plant breeding in particular. The authors are confident that ML may be a step changer for conducting research to characterize the network genetic relationship of starch genes and their modifiers, the interpretation of their genes, and environmental interaction to improve starch quality and quantity that meet the demands of breeders and end-users.

## Figures and Tables

**Figure 1 plants-11-00092-f001:**
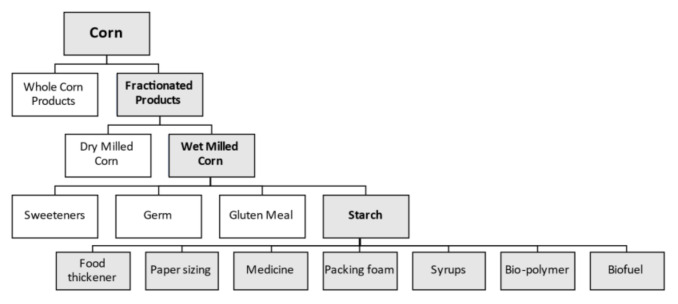
Non-food uses of corn starch derived from wet-milling process.

**Figure 2 plants-11-00092-f002:**
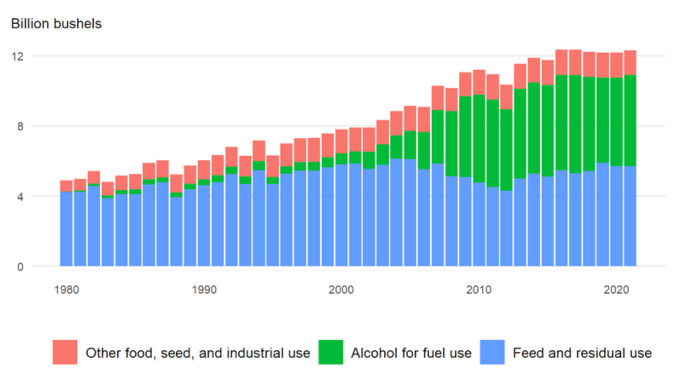
US domestic corn use (Sources: USDA national agricultural statistics service, updated in July 2021: http://www.nass.usda.gov/, accessed on 27 November 2021).

**Figure 3 plants-11-00092-f003:**
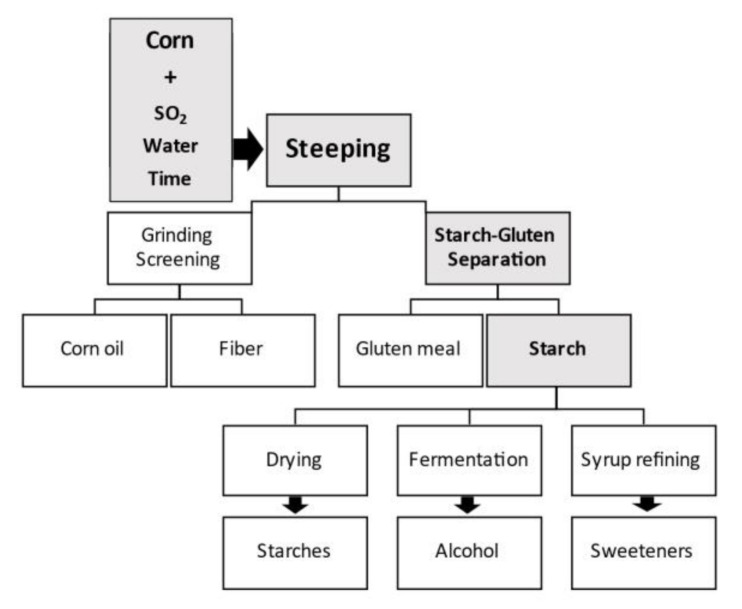
Diagram for corn wet-milling process flow to extract corn starch and output products.

**Figure 4 plants-11-00092-f004:**
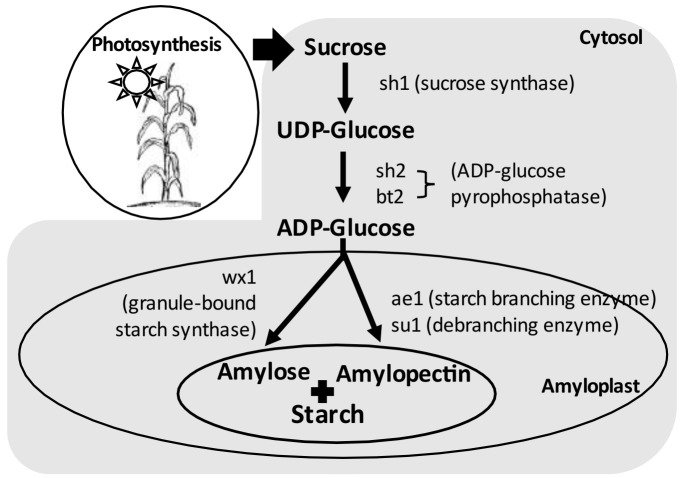
Diagram of starch biosynthesis pathway.

**Table 1 plants-11-00092-t001:** The list of corn mutations effect in starch production and their functionalities in biosynthesis.

Mutant	Gene	Enzyme	Starch Change	
			Quality	Quantity
shrunken2 (*sh2*)	*AGPLSU*	AGP large subunit		Yes
brittle2 (*bt2*)	*AGPSSU*	AGP small subunit		Yes
waxy (*wx1*)	*GBSSI*	Granule-bound starch synthase	Low amylose	
amylose extender (*ae*)	*SBEIIb*	Starch branching enzyme	High amylose	
sugary1 (*su1*)	*ISA*	Isoamylase-type debranching enzyme	Granule number and form	

## Data Availability

Not applicable.

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
