# Peer review of "Corn Starch: Quality and Quantity Improvement for Industrial Uses"

_plants, 2021, doi:10.3390/plants11010092_

Round 1

Reviewer 1 Report

The work is well written, very concise. I  appreciated the industrial cut that was given to this minireview, which is very useful for all the stakeholders. 

Minor revisions suggested: The name of the genes and the of the alleles should be written in italics and I would remove "future" from perspectives, because they are obviously future.

Author Response

Thanks so much for your valuable comment.

I revised the manuscript according to your feedback. 

Reviewer 2 Report

This review is very interesting and describes the methods to improve starch quality by using mutations of genes influencing the amylose and amylopectin biosintesis and  for industrial purposes and deals with the new challenges to increase starch quantity. To realize this last purpose, which involve several large scale and multi-dimensional datasets, the learning machine could play a pivotal role. To my opinion this review could be published after the few revisions reported below:

please add figure captions

please add table caption

write in italic the name of genes

figure 2 is grainy please enhance image resolution 

Author Response

Thanks for your valuable comment.

I revised the manuscript according to your feedback.

  1. added figure caption
  2. added table caption
  3.  correct gene names with italic letters
  4.  increased resolution of figure 2

Author Response

Thanks so much for your valuablr comment.

I revised the manuscript according to your feedback.

  1. fixed reference insertion in the text
  2. changed gene name BE to SEB (starch branching enzyme)
  3.  added title for figures and table
  4.  increased resolution for figure 2 and 4
  5.  changed abbreviated journal name i italic

Round 2

Reviewer 2 Report

Authors followed all suggestions and so to my opinion this manuscript is suited to be published in Plants journal